# Injury Symmetry in Judo

Wiesław Błach [1], Łukasz Rydzik [2,*], Arkadiusz Stanula [3], Wojciech J. Cynarski [4] and Tadeusz Ambroży [2]

1   Faculty of Physical Education & Sport, University School of Physical Education, 51-612 Wroclaw, Poland
2   Institute of Sport Sciences, University of Physical Education in Kraków, 31-571 Kraków, Poland
3   Institute of Sport Sciences, Jerzy Kukuczka Academy of Physical Education, Mikołowska 72a, 40-065 Katowice, Poland
4   Institute of Physical Culture Studies, College of Medical Sciences, University of Rzeszów, 35-959 Rzeszów, Poland
*   Correspondence: lukasz.rydzik@awf.krakow.pl

**Abstract:** Background: Each combat sport carries different risks of injury due to the specifics of the sport (including the weight categories and sex) and the fighting techniques used according to different sports regulations. The purpose of this study is to examine injury symmetry in judo. Methods: Injuries recorded in 195 people (93 women, 102 men) suffered during top-level judo tournaments were verified. Using the European Judo Union medical questionnaire, information on injuries was obtained from each injured athlete. Results: Based on the analysis of the results, it can be concluded that injuries are almost evenly distributed on the left and right sides of the body in both men and women. In women, there were 129 injuries to the left side (41.2%), 134 injuries to the right (42.8%), and 50 (16.0%) to the middle part of the body. Conclusions: injuries in judo are evenly distributed and slightly more common on the right side. Injuries occur at a greater rate during defending maneuvers than attacking maneuvers in judo. Identification and monitoring of who (tori or uke) and which side of the body sustains an injury are crucial and important in injury prevention. This knowledge makes it possible to modify existing sports regulations by eliminating behaviors (e.g., certain types of defense) to improve the safety of athletes participating in top-level competitions and the training process in terms of ensuring safety in both attack and defense.

**Keywords:** judo; sports injury; symmetry; uke; tori

## 1. Introduction

Combat sports require athletes to have all-round motor involvement and perfect mastery of the technique [1,2]. Despite the improving level of preparation of athletes for fights, there is still a risk of injury. It stems from the nature of competition, which involves a direct physical confrontation between two equal opponents [3–5]. The risk of injury applies to both the attacking athletes (collision with an opponent, an accidental kick or punch, muscle strain during lifting or pulling, various types of entanglement with an opponent's body or clothing, etc.) and the defending athletes (muscle strain during defense attempts, improper falls, etc.) [6,7]. Scientific evidence shows that, on average, Olympic combat sports athletes sustain one injury every 2.1 h of competition [5].

Each combat sport carries different risks of injury due to the specifics of the sport (including the weight categories and sex) and the fighting techniques used according to different sports regulations [8–13]. This also applies to judo and the specifics of this kind of fighting, with an emphasis on the use of throws and grappling during groundwork (*ne-waza*) [14,15]. Remarkably, in the multitude of scientific publications devoted to judo, the problem of injury rates is still relatively rarely addressed.

Body symmetry or asymmetry may be affected by load and injuries [16]. It cannot be ruled out that an athlete who performs techniques unilaterally is more likely to suffer injuries, at least due to overloading. There is no doubt that, similar to wrestling, sidedness

has an impact on tactics and fighting techniques [17,18]. The study of the lateralization of injuries in judo is important because determining the side of injury helps to identify technical factors affecting the risk of injury.

Therefore, the aim of the present study is to identify the planes of injuries with consideration for sidedness and to indicate the division in relation to the injury suffered by the attacking (*tori*) and defending (*uke*) athletes. According to sex, judo techniques can be divided into three groups: throws (nage-waza), holds (katame-waza), and strikes (atame-waza). In sports judo, only two groups of techniques are used during the fight. On the other hand, the single match itself requires appropriate offensive behavior (leaning, entering the throw, feints, and combinations) and defensive behavior (avoiding the throw, counterattack, and appropriate fall). Both offensive and defensive activities carry the risk of injury [19].

## 2. Material and Methods

### 2.1. Study Participants

We collected data from a group of 26,862 elite judokas (15,571 males and 11,291 females). They were aged between 19 and 35 years, from all judo weight categories, and had competed in 128 international tournaments of the European Judo Union (EJU) in 2005–2020, including European Judo Championships. The athletes were informed about the protocol and procedure related to the EJU injury registration form and signed informed consent forms. The EJU injury registration form was approved by the EJU Medical Commission.

Injuries recorded in 195 people (93 women, 102 men) suffered during top-level judo tournaments were verified. The inclusion criterion for the study was that the injury was recorded in the records of the European Judo Union (EJU). EJU medical data from 128 international tournaments held under the auspices of EJU were used as research material. All injuries were diagnosed by a sports physician. The experiment was approved by the Bioethics Committee at the Regional Medical Chamber (No. 287/KBL/OIL/2020).

### 2.2. Study Design

Using the European Judo Union medical questionnaire, information on injuries was obtained from each injured athlete. Injuries were divided into common, minor ones, i.e., nose bleeds, minor bruises, and abrasions, which happen commonly and do not affect the course of the fight, and serious injuries that require medical intervention and prevent the fight from continuing. A select group of fighters was chosen using the appropriate inclusion criteria, which were serious injuries that were cataloged. In addition, the present study has analyzed injuries occurring in attacking and defending athletes with respect to gender and place of injury in all the subjects studied.

### 2.3. Statistical Analysis

The data collected on the basis of the questionnaire, expressed in binary form (0 did not occur; 1 occurred), were statistically processed by calculating the frequency and percentage of the occurrence of a given variable. Significance tests for multiple comparisons of proportions based on chi-square statistics were used to calculate the differences between the analyzed variables. An alpha level of $p < 0.05$ was set as the minimal level of significance. All analyses were performed using the computer software Statistica version 13.3 (TIBCO Software Inc., Palo Alto, CA, USA).

## 3. Results

Based on the results shown in Table 1, it can be observed that in both men and women, defending athletes (*uke*) suffered far more injuries. Of the total number of women surveyed (93), *uke* suffered more than twice as many injuries (68 athletes, 69.9%). The difference of 37 (39.8%) in injuries sustained by female *tori* and *uke* was statistically significant ($p < 0.001$). In all the men included in the analysis, 72.6% of all injuries suffered were reported by

*uke*. The difference of 45 (45.1%) in injuries suffered by male *tori* and *uke* was statistically significant ($p < 0.001$).

**Table 1.** The number and percentage of injuries divided into *tori* and *uke*.

| Sex | Tori | Uke | Difference | *p*-Value |
|---|---|---|---|---|
| Women and men in total | 56 (28.7%) | 139 (71.3%) | 83 (42.6%) | $p < 0.001$ |
| Women | 28 (30.1%) | 65 (69.9%) | 37 (39.8%) | $p < 0.001$ |
| Men | 28 (27.5%) | 74 (72.6%) | 46 (45.1%) | $p < 0.001$ |

Table 2 shows the injuries divided into the side of the body on which the injury occurs. Based on the analysis of the results, it can be concluded that injuries were almost evenly distributed on the left and right sides of the body in both men and women. In women, there were 129 injuries to the left side (41.2%), 134 injuries to the right (42.8%), and 50 (16.0%) to the middle part of the body. The differences in injuries between the right and left sides, accounting for only 1.6%, were not statistically significant. Both the numbers of injuries recorded on the left and those sustained on the right side were significantly higher compared to the number of injuries recorded in the middle part of the body (by 25.2%, $p = 0.001$, and 26.8%, $p < 0.001$, respectively). Furthermore, in men, the difference in the number of injuries sustained to the left side (155, 42.1%) and the right side (158, 42.9%) was negligible, at just 0.8% ($p = 0.885$). Additionally, in men, statistically significant differences were observed in the number of injuries between the left side of the body and the middle part ($\Delta = 100$ (27.2%), $p < 0.001$) and between the right side of the body and the middle part ($\Delta = 103$ (28.0%), $p < 0.001$).

**Table 2.** Differences in injuries between sides of the body (women and men).

| Injuries | Left 1 | Right 2 | Midline 3 | Difference (1–2) | Difference (1–3) | Difference (2–3) |
|---|---|---|---|---|---|---|
| Women and men in total | 284 (41.7%) | 292 (42.9%) | 105 (15.4%) | −8 (−1.2%) $p = 0.774$ | 179 (26.3%) $p < 0.001$ | 187 (27.5%) $p < 0.001$ |
| Women | 129 (41.2%) | 134 (42.8%) | 50 (16.0%) | −5 (−1.6%) $p = 0.793$ | 79 (25.2%) $p = 0.001$ | 84 (26.8%) $p < 0.001$ |
| Men | 155 (42.1%) | 158 (42.9%) | 55 (15.0%) | −3 (−0.8%) $p = 0.885$ | 100 (27.2%) $p < 0.001$ | 103 (28.0%) $p < 0.001$ |

Table 3 shows the injuries sustained to the different parts of the body by side of injury (Table 3).

**Table 3.** Injuries suffered by athletes by body part and side.

| Injuries | Left 1 | Right 2 | Midline 3 | Difference (1–2) | Difference (1–3) | Difference (2–3) |
|---|---|---|---|---|---|---|
| Skull | 8 (2.8%) | 4 (1.4%) | 21 (20%) | 4 (1.5%) | −13 (−17.2%) | −17 (−18.6%) |
| Face | 13 (4.6%) | 8 (2.7%) | 3 (2.9%) | 5 (1.8%) | 10 (1.7%) | 5 (−0.1%) |
| Eye | 2 (0.7%) | 7 (2.4%) | 0 (0%) | −5 (−1.7%) | 2 (0.7%) | 7 (2.4%) |
| Ear | 3 (1.1%) | 1 (0.3%) | 0 (0%) | 2 (0.7%) | 3 (1.1%) | 1 (0.3%) |
| Nose | 0 (0%) | 1 (0.3%) | 9 (8.6%) | −1 (−0.3%) | −9 (−8.6%) | −8 (−8.2%) |
| Mouth | 1 (0.4%) | 1 (0.3%) | 5 (4.8%) | 0 (0%) | −4 (−4.4%) | −4 (−4.4%) |
| Neck | 4 (1.4%) | 4 (1.4%) | 49 (46.7%) | 0 (0%) | −45 (−45.3%) | −45 (−45.3%) |
| Throat | 1 (0.4%) | 1 (0.3%) | 9 (8.6%) | 0 (0%) | −8 (−8.2%) | −8 (−8.2%) |
| Clavicle/AC | 24 (8.5%) | 18 (6.2%) | 1 (1.0%) | 6 (2.3%) | 23 (7.5%) | 17 (5.2%) |
| Shoulder/Arm | 36 (12.7%) | 33 (11.3%) | 0 (0%) | 3 (1.4%) | 36 (12.7%) | 33 (11.3%) |
| Elbow joint | 49 (17.3%) | 50 (17.1%) | 0 (0%) | −1 (0.1%) | 49 (17.3%) | 50 (17.1%) |
| Forearm | 4 (1.4%) | 2 (0.7%) | 0 (0%) | 2 (0.7%) | 4 (1.4%) | 2 (0.7%) |

**Table 3.** *Cont.*

| Injuries | Left 1 | Right 2 | Midline 3 | Difference (1–2) | Difference (1–3) | Difference (2–3) |
|---|---|---|---|---|---|---|
| Wrist | 8 (2.8%) | 7 (2.4%) | 0 (0%) | 1 (0.4%) | 8 (2.8%) | 7 (2.4%) |
| Hand/Finger | 22 (7.8%) | 21 (7.2%) | 0 (0%) | 1 (0.6%) | 22 (7.8%) | 21 (7.2%) |
| Thorax | 3 (1.1%) | 17 (5.8%) | 4 (3.8%) | −14 (−4.8%) | −1 (−2.8%) | 13 (2.0%) |
| Back | 5 (1.8%) | 4 (1.4%) | 2 (1.9%) | 1 (0.4%) | 3 (−0.1%) | 2 (−0.5%) |
| Abdomen | 0 (0%) | 2 (0.7%) | 2 (1.9%) | −2 (−0.7%) | −2 (−1.9%) | 0 (−1.2%) |
| Pelvis | 1 (0.4%) | 2 (0.7%) | 0 (0%) | −1 (−0.3%) | 1 (0.4%) | 2 (0.7%) |
| Femur | 8 (2.8%) | 9 (3.1%) | 0 (0%) | −1 (−0.3%) | 8 (2.8%) | 9 (3.1%) |
| Knee | 65 (22.9%) | 55 (18.8%) | 0 (0%) | 10 (4.1%) | 65 (22.9%) | 55 (18.8%) |
| Leg | 5 (1.8%) | 8 (2.7%) | 0 (0%) | −3 (−1%) | 5 (1.8%) | 8 (2.7%) |
| Ankle | 15 (5.3%) | 23 (7.9%) | 0 (0%) | −8 (−2.6%) | 15 (5.3%) | 23 (7.9%) |
| Foot | 7 (2.5%) | 12 (4.1%) | 0 (0%) | −5 (−1.7%) | 7 (2.5%) | 12 (4.1%) |
| Other | 0 (0%) | 2 (0.7%) | 0 (0%) | −2 (−0.7%) | 0 (0%) | 2 (0.7%) |
| Total | 284 (41.7%) | 292 (42.9%) | 105 (15.4%) | −8 (−1.2%) | 179 (26.3%) | 187 (27.5%) |

## 4. Discussion

With the dynamic changes in the planes of competition in judo, the situational unbalancing of the opponent and unassisted and uncontrolled falls are often observed [11,20]. Consequently, *uke* is more likely to suffer injury than *tori*, who is usually in a more stable position when making a throw. This was confirmed by our research. The technique of falling (*ukemi*), which is constantly improved in the training process, may depend on many factors [21]. It is influenced by fatigue, the warming up of the body, the course of the fight, random situations, and the current functional and mental status of the athlete [19,21,22]. Since points are mostly awarded for the opponent's falls on their back, another factor contributing to injuries sustained by a thrown athlete is attempts to change the trajectory of the flight during the fall so as to fall on the abdomen, which guarantees no loss of points. A statistically significant difference may also be affected by the physical or technical advantage of the attacking player, which also affects the manner, form, and impact of the fall [15,23].

Statistical analysis showed that, by far, more injuries were suffered by defensive athletes (*uke*). This may be due to the fact that in judo, as argued by Bujak [24], injuries occur most often when falling and during throws. *Uke*, who defends himself or herself from falling on their back and from thus losing a point, may try to perform defensive actions at all costs, which may result in injury to body parts [21,25]. In the years in which the study was conducted, *uke* actions that were observed during judo competitions were often aimed at defending from falling on the back by supporting the body with both upper limbs (placing their hands on the mat). Such *uke* behavior has repeatedly led to serious injuries to the shoulder, elbow, and wrist joints, as confirmed by this study. Therefore, after the Tokyo Olympics, this type of defending the fall is forbidden in the IJF sports regulations and punished by the judges with a *shido* penalty, and, at the same time, a *waza-ari* is awarded despite the fact that there was no fall on the back [26]. This is one of several newly introduced regulations to prevent injuries in competition fights. Another new sports regulation to prevent serious injuries to the head and cervical spine is the prohibition of the use of defense to prevent falling on the back by using the head "bridge". This type of defense is punishable by the disqualification of the player performing such a defensive action [27]. Furthermore, the attacking athlete is punished by disqualification for the action of diving head-first on the mat while performing a forward rotation throw (e.g., *uchi-mata*, *sode tsurikomi goshi*). All these changes in the rules are aimed at minimizing the risk of injury and consequently improving the safety of fighters during a fight. Therefore, temporal and spatial analyses of injuries and their regular monitoring are extremely important and needed. Ruddy et al. [28] and Bittencourt et al. [29] found that injuries are suffered due to complex and non-linear interactions between multiple factors. Furthermore, Bahr and Holme [30] claimed that even a single factor can strongly impact the occurrence of injuries

and could show enough information to predict the injuries but only at the individual level. With this information, we can suppose that injuries may be due to technique-related factors, such as performing a specific movement structure, a different phase [31], and the spatial position of the body during a judo fight.

Analysis of the sides of the body affected by the injury, that is, the symmetry or asymmetry of injuries, reveals that in both men and women, more injuries occur to the right side of the body. This is related to the preferred side to which the throw is performed. Performing a throw (e.g., *seoi nage, uchi-mata*) to the right side or left side causes a fall on the right or left side as well, which may involve an injury on either side of the body [11]. The tendency to suffer injury on the lateral sides of the body may be due to the *uke*'s desire to fall on the side of the body, which results from preventing a fall on the back. A detailed distribution of the sides on which injuries were reported showed, in both men and women, that the number of injuries sustained to the left or right sides was significantly higher compared to the number of injuries recorded in the middle part of the body. This is certainly related to the lateralization of the athletes' bodies, i.e., performing technical elements to the preferred (right or left) side, which, in turn, engages and thus significantly exposes the lower and upper limbs on both sides to injuries [17]. This is confirmed by the number of recorded injuries to knees, elbows, shoulders, and ankle joints, both left and right.

## 5. Conclusions

This paper presents a characterization of injuries sustained by the attacking and defending athletes in judo, taking into account the side of the body that was injured. Comprehensive knowledge of the risk of injury while practicing judo and the associated risk factors provides an essential foundation for the effective development of injury prevention strategies. Identification and monitoring of who (*tori* or *uke*) and which side of the body sustains an injury is crucial and important in injury prevention. This knowledge makes it possible to modify existing sports regulations by eliminating behaviors (e.g., certain types of defense) to improve the safety of athletes participating in top-level competitions and the training process in terms of ensuring safety in both attack and defense. On the other hand, the detailed results of our research indicated that:

(1) Injuries in judo are evenly distributed on the left and right sides of the body among both male and female competitors. A more detailed analysis shows that it is slightly more common to injure the right side of the body. (2) *Uke* defenders are more likely to sustain injuries.

**Author Contributions:** Conceptualization, W.B., Ł.R. and T.A.; methodology, W.B., Ł.R. and A.S.; formal analysis, W.B., Ł.R., W.J.C., A.S. and T.A.; investigation, W.B., Ł.R. and T.A.; resources, W.B., Ł.R. and A.S.; data curation, W.B., Ł.R. and A.S.; writing—original draft preparation, W.B., W.J.C., Ł.R. and T.A.; writing—review and editing, W.B., W.J.C., Ł.R. and T.A.; visualization, W.B., Ł.R., T.A. and W.J.C.; supervision, W.B.; project administration, W.B.; funding acquisition, W.B. and T.A. All authors have read and agreed to the published version of the manuscript.

**Funding:** This research received no external funding.

**Institutional Review Board Statement:** The study was conducted according to the guidelines of the Declaration of Helsinki and approved by the Ethics Committee of the Regional Medical Board in Kraków (approval No. 287/KBL/OIL/2020).

**Data Availability Statement:** The data presented in this study are available upon request from the corresponding author.

**Acknowledgments:** A special thanks to all the members of the EJU Medical Commission, past and present, for all their efforts in the collection of all the data and injury registration forms.

**Conflicts of Interest:** The authors declare no conflict of interest.

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
