# Peer review of "Injury Symmetry in Judo"

_symmetry, doi:10.3390/sym15010013_

Round 1

Reviewer 1 Report

   This paper provides a survey of injuries in elite judo athletes. I did not understand why it was necessary to investigate the laterality of injuries in judo. The author would have to indicate in the introduction that simply investigating the occurrence of injuries is not enough and that it is important to also investigate side properties. The author states the following in the conclusion.

 “Identification and monitoring of who (tori or uke), and on which side of the body, sustains an injury is crucial and important in injury prevention.”

    However, it merely states 'important in injury prevention' and it is not clear why it is important. What can be said from the following two points mentioned in the conclusion?

1. Injuries in judo most often occur to the right side

2. Uke defenders are at more likely to sustain the risk of injury

   This article is submitted as an Article, not as materials or data, so please make your originality clear based on scientific evidence.

   The author also mentions the relationship between physical characteristics and injury in the paper, but does not include data on physical characteristics in this study. Further, the description of the methodology is insufficient. There is insufficient information in the paper, such as details of data sampling, data collection and statistical analysis methods. Also in the abstract, there is no description of the methods. The results are based on statistical processing, but since the method of statistical processing is not clear, it is impossible to judge whether the results are correct or not. The title of the paper should be revised so that the reader can understand the content of the paper appropriately.

   I cannot recommend this article for publication because it is insufficiently described. Please refer to previous similar papers and review the contents that should be included in your paper.

Author Response

Dear Reviewer,

Thank you very much for your time and valuable comments, which all have been considered and incorporated. The detailed list of responses is given below. We hope that the modifications and explanation will be acceptable for you.

Yours sincerely,

Rydzik, corresponding author

This paper provides a survey of injuries in elite judo athletes. I did not understand why it was necessary to investigate the laterality of injuries in judo. The author would have to indicate in the introduction that simply investigating the occurrence of injuries is not enough and that it is important to also investigate side properties. The author states the following in the conclusion.

 “Identification and monitoring of who (tori or uke), and on which side of the body, sustains an injury is crucial and important in injury prevention.”

A: This has been corrected

    However, it merely states 'important in injury prevention' and it is not clear why it is important. What can be said from the following two points mentioned in the conclusion?

  1. Injuries in judo most often occur to the right side
  2. Uke defenders are at more likely to sustain the risk of injury

   This article is submitted as an Article, not as materials or data, so please make your originality clear based on scientific evidence.

A: This has been corrected

   The author also mentions the relationship between physical characteristics and injury in the paper, but does not include data on physical characteristics in this study. Further, the description of the methodology is insufficient. There is insufficient information in the paper, such as details of data sampling, data collection and statistical analysis methods. Also in the abstract, there is no description of the methods. The results are based on statistical processing, but since the method of statistical processing is not clear, it is impossible to judge whether the results are correct or not. The title of the paper should be revised so that the reader can understand the content of the paper appropriately.

 A: This has been corrected

   I cannot recommend this article for publication because it is insufficiently described. Please refer to previous similar papers and review the contents that should be included in your paper.

A: Dear reviewer, thank you for your pertinent comments, which we have tried to take into account. Our paper was written based on similar scientific articles

Reviewer 2 Report

Injury symmetry in judo

In general, I recommend this manuscript for publication – in MINOR REVISION

Below, important general points to be clarified or answered:

Abstract:

Superficial conclusion about the results. Please clarify the findings 

Introduction:

Talk more about the judo modality in a more specific way

Line 45 – Explain what torie and uke movements are

Materials and methods

Line 48 - Study Design repeat last paragraph of introduction, remove or rephrase

Study participants – Congratulations on the number of athletes evaluated, but it surprises me to mention the injuries in the modality, but when evaluating 26,862 athletes, only 195 had a diagnosed injury, which gives a risk of less than 1% of injury, is that right? I would like injury rates to be better described or explained to agree with this sample

Results

Why in table 1 in the row of women the value of p is at p=0.0004 or p<0.001?

Table 3 - Data 29 (NONE) does not need to appear

Table 3 – What is the relationship between the data presented and the mentioned movements (torie and uke)?

Table 3 - I believe that the results could better explore the data presented in the table

Discussion

Line 120 – When talking about judo in the introduction explore the terms used in the article, in this case kusushi

Author Response

Dear Reviewer,

Thank you very much for your time and valuable comments, which all have been considered and incorporated. The detailed list of responses is given below. We hope that the modifications and explanation will be acceptable for you.

Yours sincerely,

Rydzik, corresponding author

Injury symmetry in judo

In general, I recommend this manuscript for publication – in MINOR REVISION

A: Thank you

Below, important general points to be clarified or answered:

Abstract:

Superficial conclusion about the results. Please clarify the findings 

 A: This has been corrected

Introduction:

Talk more about the judo modality in a more specific way

Line 45 – Explain what torie and uke movements are

 A: This has been corrected

Materials and methods

Line 48 - Study Design repeat last paragraph of introduction, remove or rephrase

 A: This has been corrected

Study participants – Congratulations on the number of athletes evaluated, but it surprises me to mention the injuries in the modality, but when evaluating 26,862 athletes, only 195 had a diagnosed injury, which gives a risk of less than 1% of injury, is that right? I would like injury rates to be better described or explained to agree with this sample

 A: This has been corrected

Results

Why in table 1 in the row of women the value of p is at p=0.0004 or p<0.001?

Table 3 - Data 29 (NONE) does not need to appear

 A: This has been corrected

Table 3 – What is the relationship between the data presented and the mentioned movements (torie and uke)?

Table 3 - I believe that the results could better explore the data presented in the table

 A: This has been corrected

Discussion

Line 120 – When talking about judo in the introduction explore the terms used in the article, in this case kusushi

 A: This has been corrected

Reviewer 3 Report

The manuscript describes the injury rates for male vs. female athletes, while defending vs. attacking, and also looks at the side (left vs. right) of injuries.

Line 19, conclusion to the abstract: “Injuries in judo most often occur to the right side”. This contradicts what was presented in the results of the abstract, where it is stated “injuries are almost evenly distributed on the left and right sides of the body” (line 16)

Abstract, line 19: “Uke defenders are at higher risk of injury.” It is unclear what this conclusion is based on (some results on this need to be presented in the abstract). A clearer conclusion could be “Injuries occur at a greater rate during defending than attacking maneuvers in Judo”.

Line 68: The statistics section is difficult to evaluate because it is written in vague terms. Please try to clarify this section.

Line 80: the wording “In contrast” does not seem to be appropriate here because you go on to describe almost identical injury rates in men.

Table 1, third column, second row: “p=0.0004 or p<0.001”. I suggest just having “p<0.001” here to be consistent with the other rows.

It is indicated that after the Tokyo Olympics, rule changes were made to reduce injuries in Judo. Was any of your data collected after the Olympics and therefore after these rule changes? Please clarify this in the manuscript so the reader can put this into context.

Conclusions, line 185: “Injuries in judo most often occur to the right side”. This contradicts your results, where you found no difference in injury rates between the left and right side.

Author Response

Dear Reviewer,

Thank you very much for your time and valuable comments, which all have been considered and incorporated. The detailed list of responses is given below. We hope that the modifications and explanation will be acceptable for you.

Yours sincerely,

Rydzik, corresponding author

The manuscript describes the injury rates for male vs. female athletes, while defending vs. attacking, and also looks at the side (left vs. right) of injuries.

  A: This has been corrected

Line 19, conclusion to the abstract: “Injuries in judo most often occur to the right side”. This contradicts what was presented in the results of the abstract, where it is stated “injuries are almost evenly distributed on the left and right sides of the body” (line 16)

  A: This has been corrected

Abstract, line 19: “Uke defenders are at higher risk of injury.” It is unclear what this conclusion is based on (some results on this need to be presented in the abstract). A clearer conclusion could be “Injuries occur at a greater rate during defending than attacking maneuvers in Judo”.

  A: This has been corrected

Line 68: The statistics section is difficult to evaluate because it is written in vague terms. Please try to clarify this section.

  A: This has been corrected

Line 80: the wording “In contrast” does not seem to be appropriate here because you go on to describe almost identical injury rates in men.

  A: This has been corrected

Table 1, third column, second row: “p=0.0004 or p<0.001”. I suggest just having “p<0.001” here to be consistent with the other rows.

  A: This has been corrected

It is indicated that after the Tokyo Olympics, rule changes were made to reduce injuries in Judo. Was any of your data collected after the Olympics and therefore after these rule changes? Please clarify this in the manuscript so the reader can put this into context.

  A: This has been corrected

Conclusions, line 185: “Injuries in judo most often occur to the right side”. This contradicts your results, where you found no difference in injury rates between the left and right side.

  A: This has been corrected

Reviewer 4 Report

The study presented for review is devoted to the problem of determining the planes of injury in martial arts, taking into account one-sidedness and the designation of the division in relation to injuries of attacking techniques and athletes who use more protective technical and tactical actions during a duel. The participants of the study were women and men - judokas of all weight categories of high class, participating in various international competitions in the age category from 19 to 35 years.

The study reliably recorded that athletes - judokas, regardless of gender, who use mainly protective technical and tactical techniques in competitions, receive significantly more different types of injuries. In turn, based on the analysis of the results carried out by the authors, it can be concluded that injuries in judokas are evenly distributed on the left and right sides of the body of athletes, both in male athletes and women athletes, however, a more detailed analysis revealed injuries in judo competition most often occur on the right side of the body.

The practical significance of this research work lies in the fact that the results of the analysis carried out by the authors will further contribute to the reduction of injuries in martial arts, which is important for increasing the mass participation of those involved in these sports. As the results of the study showed, uke is more likely to suffer injury than tori, who is usually in a more stable position when making a throw. Due to the fact that in a duel points are mostly awarded for the opponent’s falls on the back, another factor contributing to injuries sustained by a thrown athlete is attempts to change the trajectory of the flight during the fall so as to fall on the abdomen, which guarantees no loss of points.

Findings from the study makes it possible to modify existing sports regulations by eliminating behaviors to improve the safety of athletes participating in top-level competitions and the training process in terms of ensuring safety in both attack and defense.

Author Response

Dear Reviewer,

Thank you for your support and kind words. We are glad that you appreciate our hard work Yours sincerely,

Rydzik, corresponding author

The study presented for review is devoted to the problem of determining the planes of injury in martial arts, taking into account one-sidedness and the designation of the division in relation to injuries of attacking techniques and athletes who use more protective technical and tactical actions during a duel. The participants of the study were women and men - judokas of all weight categories of high class, participating in various international competitions in the age category from 19 to 35 years.

The study reliably recorded that athletes - judokas, regardless of gender, who use mainly protective technical and tactical techniques in competitions, receive significantly more different types of injuries. In turn, based on the analysis of the results carried out by the authors, it can be concluded that injuries in judokas are evenly distributed on the left and right sides of the body of athletes, both in male athletes and women athletes, however, a more detailed analysis revealed injuries in judo competition most often occur on the right side of the body.

The practical significance of this research work lies in the fact that the results of the analysis carried out by the authors will further contribute to the reduction of injuries in martial arts, which is important for increasing the mass participation of those involved in these sports. As the results of the study showed, uke is more likely to suffer injury than tori, who is usually in a more stable position when making a throw. Due to the fact that in a duel points are mostly awarded for the opponent’s falls on the back, another factor contributing to injuries sustained by a thrown athlete is attempts to change the trajectory of the flight during the fall so as to fall on the abdomen, which guarantees no loss of points.

Findings from the study makes it possible to modify existing sports regulations by eliminating behaviors to improve the safety of athletes participating in top-level competitions and the training process in terms of ensuring safety in both attack and defense.

A: Thank you for your support and kind words. We are glad that you appreciate our hard work

Round 2

Reviewer 1 Report

The authors do not seem to have responded to all the points raised. In particular, I believe that the description of the survey method in the abstract is required.

Author Response

Dear Reviewer,

Thank you very much for your time and valuable comments, which all have been considered and incorporated. The detailed list of responses is given below. We hope that the modifications and explanation will be acceptable for you.

Yours sincerely,

Rydzik, corresponding author

The authors do not seem to have responded to all the points raised. In particular, I believe that the description of the survey method in the abstract is required.

A: Thank you for the re-review, the method in the abstract has been expanded (line 16-18. Regarding the comments from the previous review, I would like to point out that everything has been improved. Changes have been made to lines 49-51, 54-60, 78-86, 89- 95, 192-206